# Vitamin D and Phenylbutyrate Supplementation Does Not Modulate Gut Derived Immune Activation in HIV-1

**DOI:** 10.3390/nu11071675

**Published:** 2019-07-21

**Authors:** Catharina Missailidis, Nikolaj Sørensen, Senait Ashenafi, Wondwossen Amogne, Endale Kassa, Amsalu Bekele, Meron Getachew, Nebiat Gebreselassie, Abraham Aseffa, Getachew Aderaye, Jan Andersson, Susanna Brighenti, Peter Bergman

**Affiliations:** 1Division of Clinical Microbiology, Department of Laboratory Medicine, Karolinska Institutet, Huddinge, 14152 Stockholm, Sweden; 2Clinical Microbiomics, 2200 Copenhagen, Denmark; 3Center for Infectious Medicine (CIM), F59, Department of Medicine Huddinge, Karolinska Institutet, Karolinska University Hospital Huddinge, 14152 Stockholm, Sweden; 4Department of Internal Medicine, Faculty of Medicine, Black Lion University Hospital and Addis Ababa University, 1176 Addis Ababa, Ethiopia; 5Armauer Hansen Research Institute (AHRI), 1005 Addis Ababa, Ethiopia; 6Department of Medicine, Division of Infectious Diseases, Karolinska University Hospital Huddinge, 14152 Stockholm, Sweden

**Keywords:** HIV-1, vitamin D, phenylbutyrate, clinical trial, TMAO, microbiota, LL-37, kynurenine/tryptophan ratio

## Abstract

Dysbiosis and a dysregulated gut immune barrier function contributes to chronic immune activation in HIV-1 infection. We investigated if nutritional supplementation with vitamin D and phenylbutyrate could improve gut-derived inflammation, selected microbial metabolites, and composition of the gut microbiota. Treatment-naïve HIV-1-infected individuals (*n* = 167) were included from a double-blind, randomized, and placebo-controlled trial of daily 5000 IU vitamin D and 500 mg phenylbutyrate for 16 weeks (Clinicaltrials.gov NCT01702974). Baseline and per-protocol plasma samples at week 16 were analysed for soluble CD14, the antimicrobial peptide LL-37, kynurenine/tryptophan-ratio, TMAO, choline, and betaine. Assessment of the gut microbiota involved 16S rRNA gene sequencing of colonic biopsies. Vitamin D + phenylbutyrate treatment significantly increased 25-hydroxyvitamin D levels (*p* < 0.001) but had no effects on sCD14, the kynurenine/tryptophan-ratio, TMAO, or choline levels. Subgroup-analyses of vitamin D insufficient subjects demonstrated a significant increase of LL-37 in the treatment group (*p* = 0.02), whereas treatment failed to significantly impact LL-37-levels in multiple regression analysis. Further, no effects on the microbiota was found in number of operational taxonomic units (*p* = 0.71), Shannon microbial diversity index (*p* = 0.82), or in principal component analyses (*p* = 0.83). Nutritional supplementation with vitamin D + phenylbutyrate did not modulate gut-derived inflammatory markers or microbial composition in treatment-naïve HIV-1 individuals with active viral replication.

## 1. Introduction

HIV-1 infection is characterized by chronic immune activation leading to accelerated ageing and disease progression [1]. A contributing factor to sustained immune activation in HIV-1 is translocation of lipopolysaccharides (LPS) and other microbial products and metabolites across a dysfunctional mucosal immune barrier in the gut [2]. Elevated plasma levels of soluble CD14 (sCD14), a marker of the monocyte response to LPS, as well as an increased kynurenine/tryptophan (kyn/trp) ratio have been linked to impaired mucosal immunity, and systemic immune activation in HIV-1 [3,4,5,6,7]. Trimethyl-N-Oxide (TMAO) is another gut microbiota-derived metabolite under investigation in HIV due to its ascribed proatherosclerotic properties [8,9,10]. A recent study found that TMAO increased the risk of atherosclerosis in HIV-1 and correlated positively with sCD14 [11]. Many observations also suggest that HIV-related disruption of the gut barrier function is accompanied by a dysregulation of the intestinal microbiota. Accordingly, an enrichment of Proteobacteria and a reciprocal reduction of Firmicutes and Bacteroidetes may sustain epithelial barrier dysfunction [3,12,13]. In particular, dysbiosis has been shown to correlate with indoleamine-2, 3-dioxygenase (IDO) mediated conversion of tryptophan to kynurenine [3]. 

Nutritional compounds with immunomodulatory properties, such as vitamin D and phenylbutyrate (PBA), play important roles in gut homeostasis [14,15,16,17,18,19] through their effects on innate and adaptive immunity [20]. One key mechanism involves vitamin D- and/or PBA-mediated induction of the antimicrobial peptide LL-37 from epithelial and immune cells [21]. LL-37 has pleiotropic protective effects that may be enhanced by vitamin D and/or PBA [22]. In the gut mucosal barrier, LL-37 production is an important innate defense mechanism that protects the host against pathogens, mainly by killing pathogenic microbes. LL-37 may also protect against HIV-1 infection via activation of autophagy that could reduce intracellular HIV replication [23], or block infection through inhibition of HIV-1 transcription [24,25,26]. Vitamin D stabilizes tight junction structures in intestinal epithelial cells [27] and appears to have regulatory effects on the intestinal microbiota composition in animal models [14,15] and in humans [28,29]. Here, recent studies demonstrate that vitamin D may modulate the relative abundance of pro-inflammatory Proteobacteria in the gut [28,29]. Of note, low 25-hydroxyvitamin D (25(OH)D) levels are highly prevalent (70%–85%) in HIV-1 infected individuals [30] and several studies have demonstrated a link between 25(OH)D deficiency and immune activation in HIV-1 [31,32]. Although there is evidence connecting vitamin D status with gut-health in HIV-1, it is still unclear whether these links are causal or a result of various confounders inherent to associative studies.

Based on these data, we hypothesised that treatment with vitamin D and PBA may modulate gut-derived immune activation, microbial composition, and subsequent production of microbial metabolites in HIV-1. We assessed these outcomes in per-protocol participants of a randomized, double-blind, and placebo-controlled trial (RCT) in Ethiopia [33]. The RCT was conducted on antiretroviral therapy (ART)-naïve HIV-1+ individuals at a time were the Ethiopian HIV-1 guidelines recommended initiation of ART first when clinical symptoms appeared and at CD4^+^ T cell counts < 350 cells/µL. The conclusion of the RCT was that vitamin D + PBA was well-tolerated and efficiently improved 25(OH)D levels but did not reduce viral load or improve CD4^+^ T cell counts [33]. Consistent with these findings we found no modulatory effects of this nutritional intervention on gut-derived inflammatory markers and metabolites, or microbial composition.

## 2. Material and Method

### 2.1. Study Design

This study was a randomized, double-blind, placebo-controlled, clinical trial conducted at the pre-ART clinic, Department of Internal Medicine, Black Lion University Hospital in Addis Ababa, between 2013 and 2015. The trial was registered on www.clinicaltrials.gov (ID: NCT01702974) on 10 October 2012 prior to inclusion of the first patient. The trial was conducted according to the principles of the Declaration of Helsinki and received ethical approval from the ethical review board in Stockholm, Sweden (EPN) and from the national, regional, and institutional review boards at the study site in Addis Ababa, Ethiopia [33].

### 2.2. Patients and Clinical Samples

ART-naïve HIV-1 infected individuals from the per-protocol cohort were included (vitamin D + PBA *n* = 81, placebo *n* = 86). All subjects provided informed written consent. Inclusion criteria were: ART-naïve HIV-positive individuals > 18 years, CD4^+^ T cells counts > 350 cells/mL, and plasma viral loads > 1000 copies/mL. Exclusion criteria were: ongoing ART or other antimicrobial drugs, antimicrobial drug treatment in the past month, hypercalcemia, pregnancy and breast-feeding, liver or renal diseases, malignancies, or treatment with cardiac glycosides. Plasma samples from peripheral blood and gut tissue biopsies (colonoscopy) were obtained from the study subjects at baseline and at follow-up after 16 weeks and stored at −85 °C until shipment and laboratory analyses in Sweden.

### 2.3. Intervention and Randomization

Patients were randomized to receive daily oral 5000 IU vitamin D and 500 mg PBA or placebo in equivalent number of tablets for 16 weeks. Vitamin D (Vigantoletten) and matching placebo tablets were donated by Merck Serono (Darmstadt, Germany); PBA and matching placebo tablets were obtained from Scandinavian Formulas (Sellersville, PA, USA). Subjects were randomized in a one-to-one allocation ratio with a selected block size of ten using computer-generated codes (Karolinska Trial Alliance, Stockholm, Sweden) [33].

### 2.4. Vitamin D

Levels of the vitamin D proform, 25(OH)D, in plasma were analyzed at the Department of Clinical Chemistry, Karolinska University Hospital, Stockholm, Sweden using a chemiluminescence immunoassay (CLIA) on a LIAISON-instrument (DiaSorin Inc., Stillwater, MN, USA), detectable range 7.5–175 nmoL/L, CV 2%–5%.

### 2.5. LL-37, sCD14 and Targeted Metabolomics

Analyses of sCD14 and LL-37 was performed at the Department of Laboratory Medicine, Clinical microbiology, Karolinska Institutet, Stockholm, Sweden, using ELISA. sCD14 (Quantikine^®^, R&D System, Abingdon, OX, UK) was measured according to the manufacturer’s recommendation. The detectable range was between 250–16,000 pg/mL, coefficient of variation (CV) 4%–7%, with a sample dilution of 1:800. LL-37 was analysed by an in-house ELISA as previously described [34], with a sample dilution of 1:200. Following methanol extraction, quantification of TMAO, choline and betaine, tryptophan, and kynurenine was performed by LC-MS/MS at the Swedish Metabolomics Centre, Umeå, Sweden as previously described [35].

### 2.6. Microbiota

16S rRNA sequencing of the microbiota found in punch biopsies from colon tissue, was performed by Clinical-Microbiomics, Copenhagen, Denmark, on a total of 38 patients, randomly selected from the treatment (*n* = 19) and placebo (*n* = 19) groups. Negative controls were introduced (using molecular grade water) and samples were bead beat using MOBIO Garnet Beads for 1 min at 1000 RPM (to release bacterial cells from the biopsy) and the supernatant bead beat with 0.1 mm glass beads for 1 min at 1000 RPM. DNA was extracted using the Qiagen AllPrep DNA/RNA/Protein Mini DNA extraction kit following the manufacturer’s protocol. The V4 region of the 16S rRNA was amplified using the universal bacterial primers 515fB and 806rB with Illumina adapters attached (35 cycles of 98 °C for 10 s, 98 °C for 30 s, 50 °C for 20 s, 72 °C for 20 s, and 72 °C for 5 min). Nextera XT indices were attached in a subsequent PCR and samples sequenced on an Illumina MiSeq desktop sequencer (Illumina) using 2 × 250 bp chemistry. The 64-bit versions of USEARCH [36] and mothur [37] was used. Paired-end reads were merged, requiring at least 30 bp overlap and a merged read length between 250 and 500 bp in length. Sequences were clustered at 97% sequence similarity using the UPARSE-OTU algorithm. Suspected chimeric operational taxonomic units (OTUs) were discarded based on comparison with the Ribosomal Database Project 6 classifier training set v9 [38] using UCHIME [39]. Taxonomic assignment of OTUs was done using the method by Wang et al. [40] with mothur’s PDS version of the RDP training database v14.

Two OTUs were found to be highly abundant, belonging to *Pseudomonas* (6%–92%) and *Halomonas* (≤8%), and had a somewhat stable ratio between them (median 16.5, IQR: 2.4, and range: 10.3–53.4). These are not known to be abundant in the gut microbiota, but better known as contaminants of laboratory and medical equipment [41,42]. It was therefore concluded that these were likely due to contamination during sampling (as the negative controls from DNA extraction and onwards showed no contamination). They were excluded in silico and following this, samples were rarified to the lowest sequence number found in a sample ≥1000 (after in silico removal of contaminating OTUs). Generalized UniFrac distances were calculated as a measure of beta diversity [43].

### 2.7. Statistical Analyses

Data were expressed as median (10–90th percentile or range) or *n* (percentage) as appropriate. Test of normality was performed by Shapiro Wilks. Accordingly, univariate comparisons between groups were performed using student *t*-test or Mann Whitney U test, and test of significance between related samples was performed by paired *t*-test or Wilcoxon signed rank test. Multiple linear regression was used to assess treatment effect adjusting for sex, age, BMI, viral load CD4, and CD4/CD8-ratio. *p* < 0.05 was considered as statistically significant. Statistical analyses were performed using statistical software IBS SSSP, version 23 (SPSS Inc., Chicago, IL, USA).

## 3. Results

### 3.1. Vitamin D + PBA Supplementation Improved Vitamin D Status in ART-Naïve HIV Patients

Baseline characteristics are presented in Table 1. The subjects were predominantly female (treatment 78%, placebo 81%), with a median age around 30 years, median viral load of 4 log 10 copies/mL, median CD4 count of around 400 cells/µL, and a decreased CD4/CD8 ratio < 0.5 (treatment 63%, placebo 57%) (Table 1). Around 70% of the subjects were vitamin D insufficient (defined as ≤ 50 nmoL/L) [44] and 6% had deficient levels (defined as < 25 nmoL/L) at baseline [44] (Table 1). 

However, 25(OH)D levels increased significantly in the treatment group from baseline to week 16 (median 37 to 120 nmoL/L, paired *t*-test: *p* < 0.001), whereas 25(OH)D remained unchanged in the placebo group (median 38 to 42 nmoL/L, paired *t*-test: *p* = 0.17) (Figure 1). By the end of the study period, vitamin D insufficiency was corrected in all but two of the HIV patients in the treatment group. Accordingly, the vitamin D + PBA group had significantly (*p* < 0.0001) higher 25(OH)D levels compared to placebo at the end of study treatment week 16, which also suggested that study compliance was high.

### 3.2. Vitamin D + PBA Supplementation Had No Effect on Immune Activation or Microbial Metabolites in Plasma Assessed in ART-Naïve HIV Patients

Next, we assessed if vitamin D and PBA supplementation had an effect on immune activation, the antimicrobial peptide LL-37, or levels of different metabolites in plasma. We found no difference in sCD14 levels between treatment and placebo (Table 2). Notably, both groups demonstrated a significant increase in LL-37 levels over time that was more pronounced in the treatment group (median 0.6 to 0.9 µg/mL, *p* < 0.001) compared to placebo (median 0.7 to 0.8 µg/mL, *p* = 0.004) (Table 2). Furthermore, no significant changes in the metabolic pathways involving kynurenine, tryptophan, kyn/trp-ratio, TMAO, or choline, could be detected in the treatment or placebo group over time (Table 2). Nor was there a significant difference between the groups at week 16 (data not shown). However, in the placebo group, betaine levels increased significantly from baseline to week 16 (median 67 to 72 µmoL/L, *p* = 0.04) (Table 2). In contrast, there was a slight reduction in betaine in the treatment group (median 73 to 69 µmoL/L, *p* = 0.09) (Table 2).

Regression analysis of the changes in outcome comparing placebo to treatment, adjusting for sex, age, body mass index (BMI), viral load CD4, and CD4/CD8-ratio, did not result in any significant effects on plasma levels of sCD14, LL-37, kynurenine, tryptophan, the kyn/trp-ratio, TMAO, or choline (Table 3). However, the decrease in betaine levels after treatment remained significant in the treatment group (*B*-11, *SE* 4, *p* = 0.007) (Table 3).

### 3.3. Vitamin D + PBA Supplementation Had No Effect on the Microbiota Composition of Gut Mucosa Assessed in ART-Naïve HIV Patients

To investigate the effect of vitamin D and PBA supplementation on the gut microbiota, we assessed 16S rRNA from mucosal gut biopsies collected at baseline and after 16 weeks of treatment (treatment *n* = 19, placebo *n* = 19). The subjects included in the microbiota analysis did not differ in demographic data from the total cohort. After removal of contamination, 23 complete sets from patients with baseline and follow-up samples remained (treatment *n* = 7, placebo *n* = 16). No significant difference was found between the placebo and treatment group when comparing the change over time (Mann-Whitney *U* tests) in the number of OTUs, (*p* = 0.71), (Figure 2a) or microbial diversity assessed by Shannon index (*p* = 0.82) (Figure 2b). Nor did paired analyses over time (Wilcoxon signed rank tests) find a significant change in number of OTUs (p_placebo_ = 0.50, p_treatment_ = 0.40) or Shannon’s diversity index (p_placebo_ = 0.46, p_treatment_ = 0.47) for either group.

Finally, there was no noticeable separation of the treatment and placebo group at baseline (Figure 3a) or week 16 (Figure 3b) in a principal component analyses based on their generalized UniFrac distances (ADONIS, *p* = 0.83).

### 3.4. Subgroup Analysis of Immune Activation and Metabolites in Vitamin D Insufficient ART-Naïve HIV Patients

Based on previous studies where increased immune-activation in HIV-1 was observed only with deficient to insufficient levels of 25(OH)D we next performed a subgroup analyses on vitamin D insufficient-subjects (≤50 nmoL/L) from the treatment (*n* = 58) and placebo (*n* = 60) arms. Treatment induced a robust increment of 25(OH)D levels (median change of 79 nmoL/L) (Table 4). Only 2 subjects remained insufficient, whereas the majority (55/58) reached optimal levels (25(OH)D > 75 nmoL/L) at week 16. Treatment did not affect the change in sCD14, kynurenine, tryptophan levels, kyn/trp–ratio or TMAO, and betaine levels. The treatment group demonstrated significantly increased levels of LL-37 (*p* = 0.02) while significantly increased levels of choline was found in plasma from the placebo group (*p* = 0.03) (Table 4). Accordingly, paired analyses (Wilcoxon signed rank tests) found a significant increase of LL-37 over time in the treatment group (median 0.69 to 0.95 µg/L, *p* < 0.001) and a significant increase of choline in the placebo group (median 53 to 61 µmoL/L, *p* < 0.001). However, regression analysis failed to demonstrate a significant treatment effect on any of the studied variables when adjusting for sex, age, BMI, viral load CD4, and CD4/CD8-ratio.

## 4. Discussion

In this study, we assessed if daily nutritional supplementation with a combination of vitamin D and PBA for 16 weeks could affect gut-derived immune activation, microbial metabolites, and modulate the gut microbiota in ART-naïve HIV-1 infected individuals. Despite significant improvement of patient’s 25(OH)D status in the treatment group, we found no clear treatment effects on sCD14 or circulating LL-37 regardless of baseline 25(OH)D levels. Likewise, there were no significant effects on plasma levels of kynurenine, tryptophan, the kyn/trp-ratio or TMAO, or choline, whereas reduced betaine levels was observed in the treatment arm. Further, our data does not support an effect of vitamin D + PBA supplementation on the microbiota of the colon.

These results are consistent with the primary analyses in the randomized vitamin D + PBA trial, showing no effects on the primary endpoints in modified intention to treat (mITT), or in per-protocol analyses including changes in plasma HIV viral load, or peripheral CD4 and CD8 T cell counts [33]. Despite a lack of effect on the primary outcomes, a potential effect on relevant markers of inflammation and microbial activation could have suggested a more long-term impact of daily nutritional supplementation with vitamin D + PBA in ART-naïve HIV patients. In inflammatory bowel disease (IBD) and chronic kidney disease (CKD), vitamin D treatment has been associated with reduced levels of systemic inflammation and disease activity [45,46,47]. However, similar to our results, vitamin D supplementation did not modulate systemic immune activation in HIV-1, or in obesity studies [48,49,50]. PBA is a histone deacetylase (HDAC) inhibitor, and a synthetic analogue to the short-chain fatty acid (SCFA) butyrate. It has been used in clinical practice for treatment of urea cycle disorders and cancer [51,52]. The rationale for combining vitamin D and PBA in this study was the synergistic effects on LL-37 production in epithelial cells and macrophages previously observed [53]. PBA treatment may also modulate mucosal inflammation and protect against bacterial invasion in murine models [54,55]. Although the aim of the combination was to strengthen the mucosal defenses in HIV-1, the combination also infers interpretational difficulties. It is possible that the observed lack of treatment-effect is the result of uncontrolled viremia opposing the modulatory effects of vitamin D and PBA through downregulating of the vitamin D receptor, as described in studies of human podocytes and T and NK cells exposed to HIV in vitro [56,57]. One may also speculate if the combination with PBA, by virtue of its role as an HDAC-inhibitor [58], may have reactivated latent HIV [59], thus confounding the desired outcomes. It can also be argued that the study design with an inclusion criteria of CD4 count > 350 cells/µL may have selected a cohort with less dysregulation of the gut immune barrier where minor modulation of inflammatory signals and microbiota were too weak to be detected in the available patient material. Furthermore, this trial was conducted at a time when ART was initiated in HIV patients with declining CD4 T cell counts, whereas present WHO guidelines recommends start of ART at HIV diagnosis regardless of CD4 counts.

The role of circulating LL-37 remains to be defined, but it has been used by others as a surrogate marker for the effect of vitamin D on innate immunity. Surprisingly, vitamin D supplementation did not increase LL-37 levels in plasma as previously described [60,61]. Other studies suggest that LL-37 levels in plasma may be indicative of persistent inflammation in patients with active pulmonary TB [62]. Assessment of LL-37 levels locally in the gut compared to blood would clarify how to properly interpret local and systemic levels of LL-37. In this study, we also failed to detect a treatment effect on the IDO1-mediated tryptophan-kynurenine pathway. IDO1 activity is stimulated by LPS and other inflammatory mediators [63], but may also be related to gut dysbiosis [3]. The unchanged levels of tryptophan, kyrurenine, and the kyn/trp-ratio suggest a lack of measurable gut immunomodulation of the vitamin D and PBA supplementation. Interestingly, a previous study reported enhanced IDO1 activity in macrophages stimulated with butyrate in vitro [64]. Although PBA has not been studied in this respect, it is possible that PBA may have a similar effect on IDO1 activity that in part could explain the lack of reduction in the kyn/trp-ratio.

Contrary to Obeid et al. [65], we did not find lowered TMAO levels or increased choline levels with vitamin D supplementation. Nor did our data support our hypothesis that TMAO levels would increase with supplementation, based on our previous observation suggesting that disturbed gut immune barrier function negatively affects TMAO levels in ART-naïve individuals with HIV-1 [66].

Choline and betaine are both TMAO precursors [9,67] involved in the methionine-homocysteine cycle. Unexpectedly, betaine levels decreased in the treatment group, whereas TMAO and choline levels remained unchanged. The relevance of this finding is unclear and further studies on this topic are needed.

Although HIV-1-infected individuals had a microbiota composition that was significantly distinct from HIV-negative individuals (ADONIS, *p* = 0.002) (data no shown), we could not detect significant differences in the microbiota comparing vitamin D + PBA to the placebo control. Bashir et al. showed that vitamin D supplementation in healthy individuals caused major microbial changes with a decreased relative abundance of Proteobacteria [29]. However, the observed microbial changes were only found in biopsies from the upper gastrointestinal (GI) tract whereas, similar to our findings, there was no effect on the microbiota in colonic biopsies, or in stool [29] suggesting a site dependent effect. It should be noted, however, that vitamin D supplementation caused a shift of the microbial composition in stool samples from patients with cystic fibrosis [28], suggesting that vitamin D mediated effects on microbiota may be different for various anatomical sites or between diseases.

This study has several strengths. First, the samples were collected from a bona fide RCT, which should minimize the risk of selection bias. Second, the adherence to treatment was excellent with a clinically relevant elevation of 25(OH)D-levels in the majority of subjects in the treatment arm, of whom most were vitamin D insufficient at baseline. There are also several limitations. First, the number of mucosal biopsies collected both at baseline and follow-up was low. Second, loss of data due to contamination and in silico removal reduce the strength of the microbial analysis. Moreover, the study lacks information on the participants’ diet, with potential impact on microbial composition and TMAO production. Finally, the study design precluded assessment of the individual effect of vitamin D and PBA treatment.

## 5. Conclusions

In this RCT, providing nutritional supplementation with vitamin D and PBA to ART-naïve HIV-1 patients, we could not detect any beneficial effects on the chosen markers of immune activation, microbial metabolites or colonic mucosal microbiota. We suggest that the lack of effect may in part be related to persistent viral replication in untreated HIV, where PBA, by virtue of its role as HDAC-inhibitor, may have contributed to reactivation of latent HIV. We propose further studies of vitamin D supplementation in HIV-infected individuals on ART to better evaluate the modulatory effects in a virally controlled setting.

## Figures and Tables

**Figure 1 nutrients-11-01675-f001:**
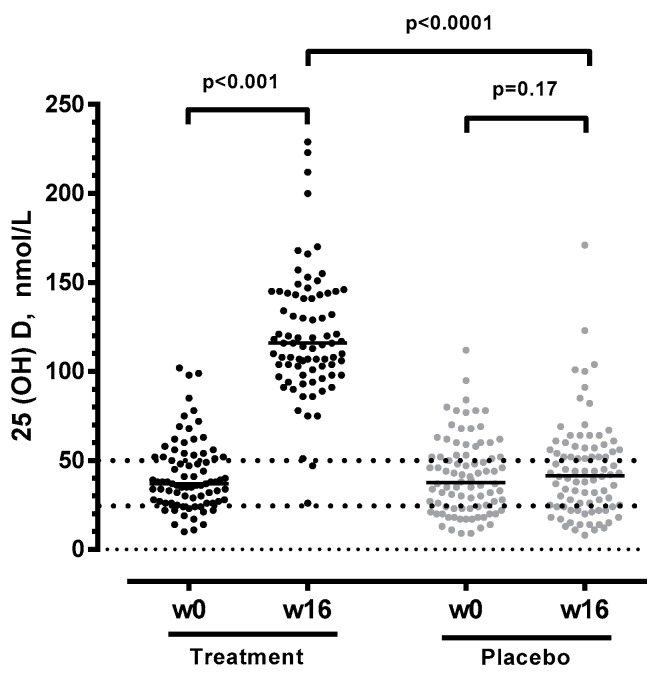
Vitamin D status assessed at baseline and at weeks 16 after daily supplementation with 5000 IU vitamin D and 500 mg phenylbutyrate (*n* = 81), or placebo (*n* = 86). *P* values are generated by Wilcoxon signed-rank and Mann-Whitney *U* test.

**Figure 2 nutrients-11-01675-f002:**
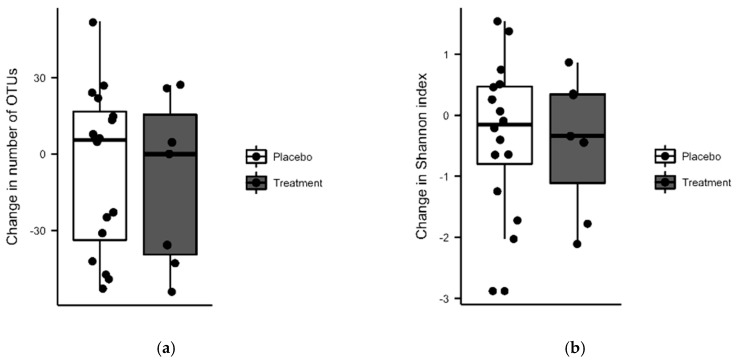
No significant treatment effects on microbiota was found in colonic biopsies from baseline and week 16 (treatment *n* = 7, placebo *n* = 16) in analyzes of number of operational taxonomic units (OTUs) (**a**), Shannon microbial diversity index (**b**).

**Figure 3 nutrients-11-01675-f003:**
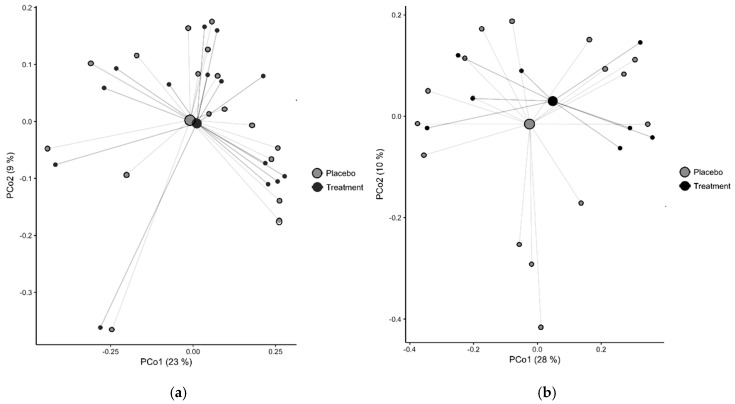
No significant treatment effects on microbiota was found in colonic biopsies (treatment *n* = 7, placebo *n* = 16) in principal component analyses at baseline (**a**) and week 16 (**b**).

**Table 1 nutrients-11-01675-t001:** Baseline characteristics of study participants.

	Treatment	Placebo	*p-*Value
Number of participants	81	86	
Female gender **	62 (78)	70 (81)	0.56
Age (years *)	32 (19–59)	30 (19–62)	0.63
Body mass index (BMI)	21 (18–27)	22 (18–28)	0.45
Viral load (log10 copies/mL)	4.0 (3.3–5.0)	3.8 (3.2–5.1)	0.20
CD4 (cells/µL)	410 (280–700)	410 (270–570)	0.37
CD4/CD8 ratio	0.42 (0.23–0.79)	0.46 (0.24–0.81)	0.85
25(OH)D_,_ (nmoL/L)	37 (22–69)	38 (17–72)	0.91
Deficient (< 25 nmoL/L) n (%)	15 (19)	25 (29)	
Insufficient (25–50 nmoL/L) n (%)	43 (53)	35 (41)	
Sufficient (> 50 nmoL/L) n (%)	23 (28)	27 (30)	

Data are expressed as median (10–90th percentile, or range *) or *n* (percentage) **. *p*-values were generated by Student *t*-test.

**Table 2 nutrients-11-01675-t002:** Treatment effect on metabolites and inflammation in antiretroviral therapy (ART)-naïve HIV-1.

	Treatment (*n* = 81)			Placebo (*n* = 86)		
	Baseline	Week 16	*p-*Value	Baseline	Week 16	*p-*Value
sCD14 (µg/mL)	1.8 (1.0–3.7)	1.9 (1.1–3.4)	0.82	1.9 (1.2–4.0)	2.2 (1.1–4.1)	0.21
LL37 (µg/mL)	0.6 (0.3–1.3)	0.9 (0.4–1.4)	<0.001	0.7 (0.4–1.2)	0.8 (0.4–1.5)	0.004
Kynurenine (µmoL/L)	2.3 (1.6–3.6)	2.5 (1.7–3.9)	0.34	2.3 (1.2–3.6)	2.5 (1.4–3.6)	0.09
Tryptophan (µmoL/L)	36 (22–51)	35 (20–53)	0.25	35 (23–49)	37 (20–53)	0.51
kyn/trp ratio	0.07 (0.04–0.14)	0.07 (0.04–0.12)	0.35	0.06 (0.04–0.12)	0.07 (0.04–0.11)	0.14
TMAO * (µmoL/L)	2.1 (0.5–5.9)	2.4 (0.9–6.3)	0.25	2.5 (0.5–7.0)	3.0 (0.8–7.0)	0.12
Choline (µmoL/L)	62 (44–76)	63 (46–79)	0.71	56 (44–72)	61 (47–78)	0.10
Betaine (µmoL/L)	73 (45–130)	69 (41–110)	0.09	67 (47–100)	72 (47–120)	0.04

Data are expressed as median (10–90th percentile). *p*-values was generated by paired *t*-test or paired Mann-Whitney *U* tests *. Soluble CD14 (sCD14), kynurenine (kyn), tryptophan (trp), and trimethyl-N-Oxide (TMAO).

**Table 3 nutrients-11-01675-t003:** Differences in outcomes between placebo and treatment arm.

Change in Dependent Variable	β	SE	*p*-Value
sCD14 (µg/mL)	−0.04	0.18	0.81
LL37 (µg/mL)	0.04	0.05	0.41
Kynurenine (µmoL/L)	−0.09	0.13	0.45
Tryptophan (µmoL/L)	0.69	2.0	0.73
kyn/trp ratio	0.12	0.12	0.33
TMAO (µmoL/L)	0.08	0.83	0.93
Choline (µmoL/L)	−2.6	2.4	0.29
Betaine (µmoL/L)	−11	4	0.007

Results presented as beta coefficients (β), standard error (SE), and corresponding *p*-values. Model adjusted for sex, age, body mass index (BMI), Viral load, CD4, CD4/CD8-ratio, and treatment with male gender and placebo as reference group.

**Table 4 nutrients-11-01675-t004:** Difference in outcomes in vitamin D insufficient individuals in treatment and placebo arm.

Change in Variable	Treatment *n* = 58	Placebo *n* = 60	*p-*Value
25(OH)D (nmoL/L)	79 (51–120)	2.0 (−9–17)	<0.001
sCD14 (µg/mL)	0.18 (−1.34–1.22)	0.05 (−1.07–1.12)	0.76
LL37 (µg/mL)	0.16 (−0.19–0.64)	0.05 (−0.34–0.32)	0.02
Kynurenine (µmoL/L)	0.05 (−0.96–1.2)	0.17 (−0.03–0.03)	0.50
Tryptophan (µmoL/L)	1.3 (−10–19)	2.2 (−14–14)	0.87
kyn/trp ratio	−0.00 (−0.03–0.03)	0.00 (−0.03–0.03)	0.28
TMAO (µmoL/L)	0.35 (−3.1–2.7)	0.27 (−2.7–5.6)	0.55
Choline (µmoL/L)	1.5 (−15–16)	9.6 (−11–19)	0.03
Betaine (µmoL/L)	−2.7 (−33–31)	3.5 (−24–33)	0.22

Data are expressed as median (10–90th percentile). *p*-values were generated by Student *t*-test.

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
