# Peer review of "Vitamin D and Phenylbutyrate Supplementation Does Not Modulate Gut Derived Immune Activation in HIV-1"

_nutrients, 2019, doi:10.3390/nu11071675_

Reviewer 1 Report

This is a very interesting trial registered with ClinicalTrals.Gov addressing an emerging question. The manuscript is well written and clearly presents the design and findings.

Introduction

Conscise and to the point. Sufficiently introduces the topic but could be ellaborate, especially about the population in under study.

The final paragraph is more like an abstract. The vast majority of this information does not belong in the Introduction.

Material and Method

The study design is not inappropriate (would have preferred factorial or cross-over).

The study design is sufficiently described to allow for replication for the most part. Somewhat lacking for the gut tissue biopsies.

Results

Figure 2C is difficult to read.

The subgroup analysis was a very good idea, as improvement of vitamin D status (from deficiency to sufficiency or even optimal) may be more important than absolute status. You do not explain that anywhere in this manuscript, however...and this is an important concept in general and in why your finding may be negative despite a potential relationship still existing.

Discussion

Your statements are a bit too absolute given the unknown of vitamin D status improvement vs. absolute status.

Reviewer 2 Report

This work by Missailidis and colleagues evaluates the potential role of vitamin D and PBA on targeted immunologic and metabolic markers as well as on gut microbiota composition in ART-naïve HIV-1 subjects. Although the paper presents mostly “negative” results, the study hypothesis was solid and the experiments well designed and executed. I have no particular comments since the authors already addressed the strengths and the limitations of the study.

I would suggest to substitute the term 16S rDNA with 16S rRNA gene sequencing and the term microbiome with microbiota since you are evaluating only the taxonomic composition of the intestinal microbial community and not its genomic content.
